# PRC2-AgeIndex as a universal biomarker of aging and rejuvenation

Mahdi Moqri [1,2,3,4,17], Andrea Cipriano [2,5,17], Daniel J. Simpson [2,5,17], Sajede Rasouli [2,5], Tara Murty[6], Tineke Anna de Jong [2,5], Daniel Nachun [7], Guilherme de Sena Brandine[8], Kejun Ying [3], Andrei Tarkhov [3], Karolina A. Aberg [9], Edwin van den Oord [9], Wanding Zhou [10], Andrew Smith[8] Crystal Mackall [6,11,12], Vadim N. Gladyshev [3], Steve Horvath [13,14], Michael P. Snyder [4,15,18] ✉ & Vittorio Sebastiano [2,5,16,18] ✉

DNA methylation (DNAm) is one of the most reliable biomarkers of aging across mammalian tissues. While the age-dependent global loss of DNAm has been well characterized, DNAm gain is less characterized. Studies have demonstrated that CpGs which gain methylation with age are enriched in Polycomb Repressive Complex 2 (PRC2) targets. However, whole-genome examination of all PRC2 targets as well as determination of the pan-tissue or tissue-specific nature of these associations is lacking. Here, we show that low-methylated regions (LMRs) which are highly bound by PRC2 in embryonic stem cells (PRC2 LMRs) gain methylation with age in all examined somatic mitotic cells. We estimated that this epigenetic change represents around 90% of the age-dependent DNAm gain genome-wide. Therefore, we propose the "PRC2-AgeIndex," defined as the average DNAm in PRC2 LMRs, as a universal biomarker of cellular aging in somatic cells which can distinguish the effect of different anti-aging interventions.

DNA methylation (DNAm) is the epigenetic modification most strongly associated with aging[1,2,3]. Numerous studies have identified examples of specific CpGs whose methylation levels are strongly correlated with age; these CpGs have been used to build epigenetic clocks[4]. Many CpG-based clock methods trained using chronological age alone suffer from limited mechanistic interpretability[5]. In addition, these epigenetic clocks use a small number of CpG sites, relative to the rest of the genome, that are not necessarily representative of the epigenetic aging profile as a whole[4,6]. Several studies have demonstrated a global increase in epigenetic entropy as a result of the loss of epigenetic

¹Department of Biomedical Data Science, School of Medicine, Stanford University, Stanford, CA, USA. ²Department of Obstetrics & Gynecology, School of Medicine, Stanford University, Stanford, CA, USA. ³Department of Medicine, Brigham and Women's Hospital, Harvard Medical School, Boston, MA, USA. ⁴Department of Genetics, School of Medicine, Stanford University, Stanford, CA, USA. ⁵Institute for Stem Cell Biology and Regenerative Medicine, School of Medicine, Stanford University, Stanford, CA, USA. ⁶Center for Cancer Cell Therapy, Stanford Cancer Institute, School of Medicine, Stanford University, Stanford, CA, USA. ⁷Department of Pathology, School of Medicine, Stanford University, Stanford, CA, USA. ⁸Quantitative and Computational Biology, University of Southern California, Los Angeles, CA, USA. ⁹Center for Biomarker Research and Precision Medicine, Virginia Commonwealth University, Richmond, VA 23298, USA. ¹⁰Center for Computational and Genomic Medicine, The Children's Hospital of Philadelphia, Philadelphia, PA, USA. ¹¹Department of Pediatrics, Division of Hematology and Oncology, School of Medicine, Stanford University, Stanford, CA, USA. ¹²Department of Medicine, Division of Stem Cell Transplantation and Cell Therapy, School of Medicine, Stanford University, Stanford, CA, USA. ¹³Department of Human Genetics, David Geffen School of Medicine, University of California, Los Angeles, CA, USA. ¹⁴Altos Labs, San Diego, CA, USA. ¹⁵Center for Genomics and Personalized Medicine, Stanford University, Stanford, CA, USA. ¹⁶Stanford Maternal & Child Health Research Institute, Stanford University, Stanford, CA, USA. ¹⁷These authors contributed equally: Mahdi Moqri, Andrea Cipriano, Daniel J. Simpson. ¹⁸These authors jointly supervised this work: Michael P. Snyder, Vittorio Sebastiano. ✉e-mail: mpsnyder@stanford.edu; vsebast@stanford.edu

information, which occurs when regions that are predominantly hypo- or hypermethylated at birth gradually transition to a state of partial methylation whereby methylation of these sites becomes less predictable[7,8]. This increase in epigenetic entropy is not observed at sites most highly correlated with age that are used in epigenetic clocks, as these sites instead transition from one methylation state to the other[9]. We have recently demonstrated that during aging, mitotic somatic cells undergo a global loss of DNAm particularly in heterochromatic gene-poor regions[10]. In the current study, we perform an extensive analysis of the age-dependent gain of DNAm.

A relatively small fraction of mammalian genomes lacks DNA methylation, best exemplified by broad CpG islands (CGI) and certain non-CGI site[11]. This fraction of the genome is known as low-methylated regions (LMRs)[12] and is also referred to as non-methylated islands[11], DNAm valleys[13], DNAm canyons[14], and DNAm nadirs[15]. LMRs are highly enriched in regulatory regions[14]. LMRs at actively transcribed promoters are proposed to remain devoid of DNAm through the formation of DNA-RNA hybrid structures (R-loops) that protect the DNA from methyltransferase activity[16]. LMRs at repressed promoters are believed to actively maintain hypomethylation through the methylcytosine dioxygenase TET1 while facilitating the binding of PRC2[17]. The biochemical antagonism between DNA methylation and PRC2 action has been well documented[18–21]. Interestingly, PRC2-bound regions have been found to be hypermethylated in the aging process and in cancer[22–29]. While several studies have identified the enrichment of PRC2 targets within the age and cancer-dependent methylomes[4,25,26,29–37], a comprehensive method that segregates samples in an age-dependent manner (i.e. by using these biologically relevant regions, rather than single CpGs) has yet to be developed. Here, we examine genome-wide age-dependent changes in DNAm by using multiple bulk and single-cell datasets of young and old samples from different human and mouse tissues analyzed by Whole-Genome Bisulfite Sequencing (WGBS), Reduced-Representation Bisulfite Sequencing (RRBS), and HumanMethylation450 BeadChip (HM450) microarrays. We then propose the PRC2-AgeIndex as a biologically informed approach to measure aging, based on the progressive gain of DNAm at LMRs which are highly bound by PRC2 (High-PRC2 LMRs) in human embryonic stem cells (hESCs) genome-wide. The PRC2-AgeIndex is assay-agnostic, can be applied to different tissues without any previous training step, is robust to site-specific variability and noise, and identifies age-dependent changes at the level of genomic regions, such as chromosomes. In contrast to our recent work on global cell-agnostic loss of DNAm with aging in heterochromatin gene-poor regions, this work identifies genome-wide gain of methylation at LMRs with aging, providing a universal and robust biomarker of aging and rejuvenation.

## Results

### Low-methylated regions are shared across different cell types
Many of the regions most commonly observed in the studies of the aging methylome, such as the ELOVL2 promoter, are located at LMRs in the genome of the tissue of interest. In addition, these same LMRs are also lowly methylated in other cell types and tissues, including human embryonic stem cells (hESCs). To systematically examine this pattern of low methylation, we analyzed WGBS data from human CD4+ T-cells (neonatal and centenarian), human epidermis from sun-exposed skin (young and old), and hESCs (see Supplementary Data 1) by identifying and comparing LMRs within all samples for each tissue type. From this analysis, we observed a significant overlap in LMRs in the 3 cell types analyzed (Fig. 1) We found that 90% of LMRs in T-cells and 67% of LMRs in the epidermis overlap ("Methods" section, Fig. 1a). In addition, 91% of the T-cell/epidermis common LMRs are also lowly methylated in hESCs (Fig. 1a). In agreement with previous studies, we observed that a subset of these regions is highly bound by PRC2 core subunits, EZH2 and SUZ12 in hESCs (see ELOVL2 promoter,

for example, Fig. 1b). Pursuing PRC2 binding in these regions of interest, we examined age-dependent changes in DNAm within all LMRs and their association with PRC2-binding levels. We observed that the majority of LMRs, both in T-cells and epidermis, which are highly bound by PRC2 in hESCs, gain DNAm by age. Specifically, among the top 1000 LMRs in T-cells and epidermis with the highest binding levels of PRC2 in hESCs, 95% and 83% are hypermethylated with age, respectively (Fig. 1c, see "Methods" section). A similar trend was also observed in mice by analyzing WGBS data from young versus old murine hepatocytes, but to a lower extent, e.g., for a lower difference in DNAm, as the age-dependent DNAm gain seems to be smaller in mice (see next section for more details). In further analyses, we focused on the top 1000 LMRs of a given tissue (ranked based on the binding of PRC2 in hESCs) (Fig. 1c and Source Data File 1) that we will refer to as "high-PRC2 LMRs". Their average size is approximately 3 kb (Supplementary Fig. 1a), and more than 90% of the CpGs harbored in them are shared among three different tissue types (CD4 T-cells, epidermis and fibroblasts) (Supplementary Fig. 1b). Interestingly, by investigating the genome categories of the top 1000 LMRs compared to the total LMRs (as background), we found them to be, as expected, enriched in CpG-rich regions (Supplementary Fig. 1c) mostly mapping on developmental genes (Supplementary Fig. 1d).

### High-PRC2 LMRs consistently gain DNA methylation by age in different tissues
Prompted by our previous findings (Fig. 1), we next examined the association of PRC2 binding in hESCs with DNA methylation during aging. Investigating throughout the genome in different tissues and across species, we compared DNAm levels within all LMRs between young and old samples. WGBS data from human CD4+ T-cells (7 samples aged 18–103 years and 1 cord blood age 0), human epidermis (12 samples from 3 young and 3 old subjects, each with sun-exposed and sun-protected skin), as well as mouse hepatocyte samples (4 samples aged 2 months and 4 samples aged 22 months) were analyzed (Fig. 2a). As our group has previously shown[10], the overall DNAm level during aging is generally decreased, with the effect being more prominent in blood than in epidermis possibly due to the higher coverage and resolution of the blood dataset. Strikingly, we observed that the high-PRC2 LMRs gain methylation with age in all the samples analyzed (Fig. 2a, right panels). This robust and consistent age-dependent gain of methylation pattern can be further appreciated at the chromosomal level (Fig. 2b). Interestingly, by comparing epidermis samples from 6 young (18–25 years) and 6 old (74–83 years) individuals, we confirmed the same trend in DNAm gain at high-PRC2 LMRs that was observed in human T-cells (Fig. 2a). We also observed that while no differences were found in young sun-exposed vs sun-protected samples, old sun-exposed epidermis samples were significantly more methylated in these regions, compared to old sun-protected epidermis samples (Fig. 2c). Thus, this phenomenon of methylation gain at high-PRC2 LMRs is not only strongly dependent on the chronological age but also on the biological age, which is in turn influenced by the environment.

### PRC2-AgeIndex is measurable using different assays
Exploring the effect of methylation gain at PRC2-bound LMRs in assays beyond WGBS, we analyzed 4518 HM450 microarray human blood samples of different ages (13–100 years old), from four large publicly available DNA methylation datasets (Supplementary Data 1). While an overall loss in DNAm was not observed given the low representation of CpGs in the arrays (1.5% of all CpGs in the genome), by ranking all low-methylated CpGs (LMCs) by PRC2 binding, we found a consistent gain of methylation at high-PRC2 LMCs with age across all four datasets (Fig. 3a). We also analyzed a large dataset of RRBS samples from 6 different tissues (adipose, kidney, muscle, lung, liver, and blood, Supplementary Data 1) from young and old mice. By comparing all low-

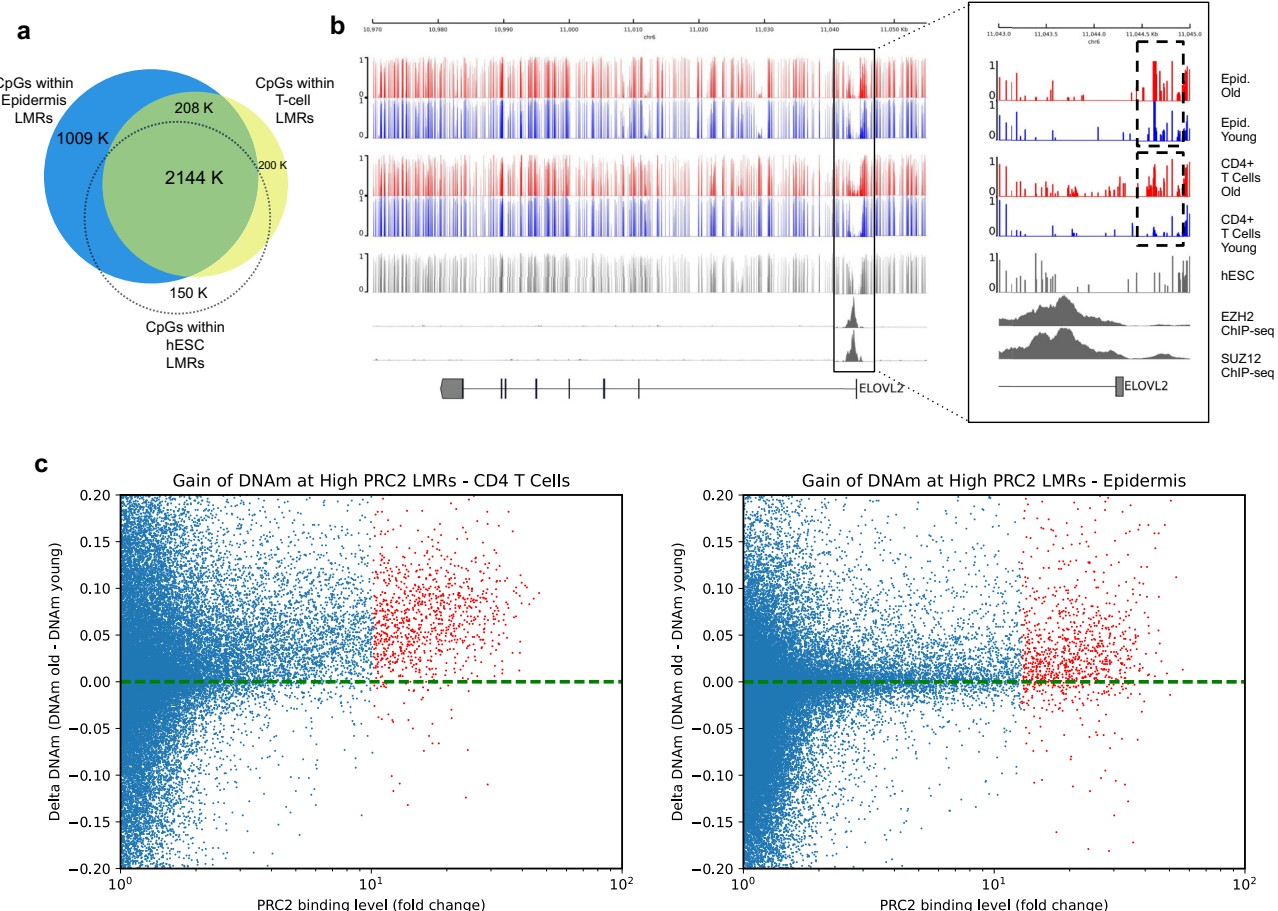

**Fig. 1 | Low-methylated regions are common across different cell types. a** Venn diagram showing the overlap between the low-methylated regions (LMRs) identified in epidermis (GSE52972), CD4+ T-cells (GSE31263), and hESCs (H1, GSE16256). **b** Genome browser visualization of ELOVL2 gene locus showing the DNA methylation tracks of old and young epidermis (GSE52972), old and young CD4+ T-cells (GSE31263), and the ChIP-Seq coverage of SUZ12 and EZH2 in hESCs from ENCODE (see "Methods" section and Supplementary Data 1). **c** Dotplot showing predominantly positive Δ DNAm (DNAm old - DNAm young) at LMRs with high-PRC2 binding (top 1000, highlighted in red) in CD4+ T-cells (left panel, mean: 0.08., variance: 0.008) and epidermis (right panel, mean: 0.05, variance: 0.007).

methylated CpGs and ordering them by PRC2 binding, we observed a consistent gain of methylation within high-PRC2 LMCs across all the tissues (Fig. 3b), suggesting that the gain of methylation at PRC2 sites with aging is evolutionarily conserved in mammals and can be captured using different DNA methylation assays. Notably, similar results were also obtained by analyzing DNAm levels at PRC2 LMRs in mouse liver by using single-cell WGBS (scWGBS) data (Fig. 3c), demonstrating the robustness of our method with noisy, low-coverage data. Since the PRC2-AgeIndex is able to discern samples similarly based on age to DNAm-based age/health predictors, a comparative analysis was conducted assessing the effect size of the PRC2-AgeIndex against over forty established DNAm-based age/health predictors, available via the Biolearn[38] and ClockBase[39] libraries. After applying these predictors to multiple WGBS datasets, the PRC2-AgeIndex exhibited a robust correlation with age, demonstrating comparable performance to other well-established age predictors (Supplementary Fig. 2).

**PRC2-AgeIndex as a biomarker of rejuvenation treatments**

Given the strong correlation between DNAm at PRC2 LMRs and physiological aging, the effects of rejuvenation interventions on PRC2-AgeIndex was interrogated. Analyzing two publicly available genome-wide methylation sequencing datasets of aging interventions in mouse liver (WGBS) and blood (RRBS), the methylation levels at the high-PRC2 LMRs for all samples were calculated. In mouse liver, old mice subjected to long-term caloric restriction (CR) had slightly lower PRC2-

AgeIndex compared to old control mice (significant in one-tailed t-test after removing outliers, p-value = 0.039, details in "Methods" section), while no significant change in the PRC2-AgeIndex was observed in mice treated with rapamycin (Fig. 4a). This suggests that the anti-aging effects of CR may be associated with amelioration of methylation gain at PRC2 LMRs as previously shown[40,41] while it is less likely that rapamycin induces epigenetic rejuvenation via the demethylation of highly methylated PRC2 LMRs. Given the small sample size and that many factors such as dosage, regimen, and age-of-onset of treatment are critical parameters for such interventions, further replication of this study using a larger cohort is needed to further explore these initial findings. When analyzing DNAm data from a larger cohort of 195 RRBS blood samples from CR and control mice, lower PRC2-AgeIndex was observed in CR mice compared to control mice[42] (Fig. 4b). To study the rejuvenation effect of epigenetic reprogramming[43–47], DNAm changes were examined in a rejuvenation treatment of mouse skin through partial reprogramming via Yamanaka Factors (OSKM)[43] in vivo, using HorvathMammalMethylChip40[48] methylation data (GSE190665). Interestingly, both fully and partially epigenetically reprogrammed samples had significantly lower PRC2-AgeIndex compared to the control samples (p-value 0.03, one-sided, independent t-test, Fig. 4c), suggesting that the age-dependent gain of methylation is reversible and can be used to assess the levels of epigenetic rejuvenation. Taken together, these data clearly demonstrate that PRC2-AgeIndex is a reliable, assay-agnostic, pan-tissue, murine, and human quantification

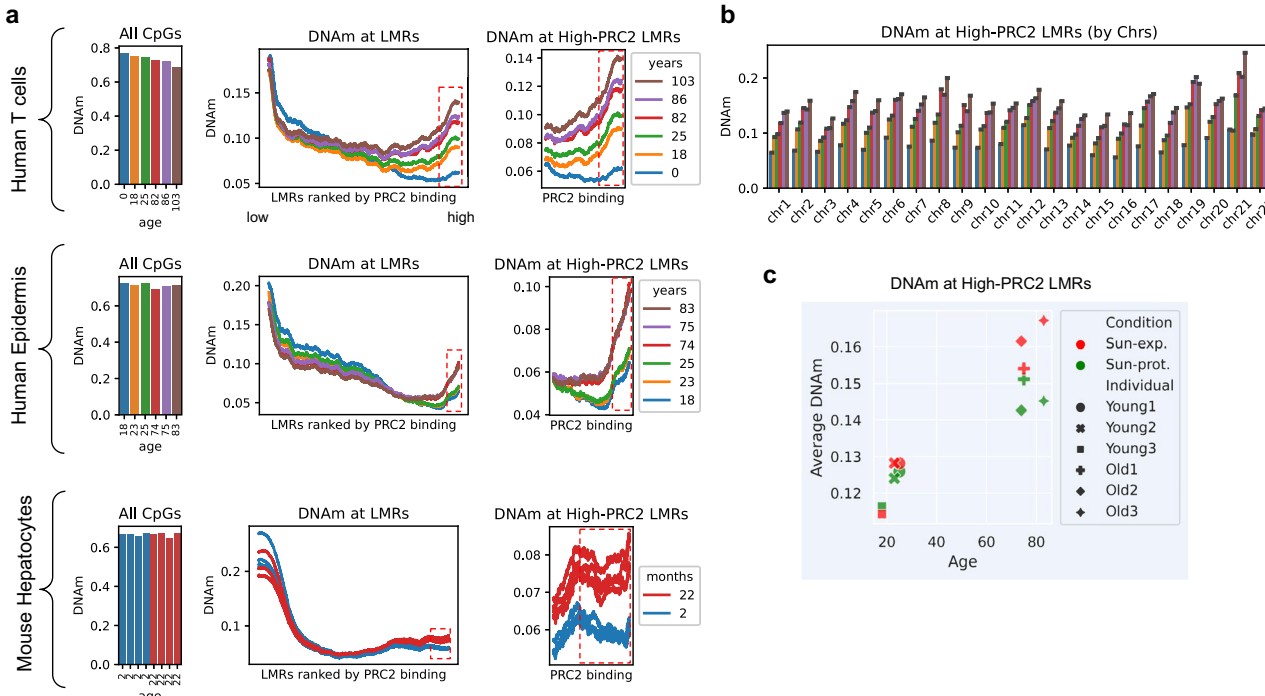

**Fig. 2 | PRC2 targets gain DNA methylation by age. a** Average DNAm levels at CpGs across the whole genome (left panel) and at the LMRs ranked by the level of PRC2 binding in hESCs or mESCs (right panels). Data were obtained by analyzing WGBS datasets from human CD4+ T-cells (6 samples 18–86 years old, GSE79798 and 2 samples 0 and 103 years old, GSE31263), human epidermis (12 samples from 6 different ages, each age including a sun-exposed and sun-protected sample averaged for this analysis, GSE52972), and mouse hepatocytes (8 samples, 4 2 months vs 4 22 months, GSE89274). PRC2 binding was evaluated using EZH2 and SUZ12 ChIP-Seq data from ENCODE in H1 hESCs and in mESCs (see Supplementary Data 1). One outlier sample was removed (see "Methods" section). **b** Average DNAm levels at high-PRC2 LMRs shown for each autosomal chromosome. **c** Average DNAm levels at high-PRC2 LMRs in skin-exposed versus skin-protected samples derived from donors of different ages. Each sample of the same age (skin protected and skin exposed) derives from the same donor. Data are from the same human epidermis WGBS datasets shown in a. High-PRC2 LMRs (top 1000 PRC2-binding LMRs in a given tissue) are highlighted with dotted red boxes.

that, in addition to measuring biological aging, can also quantify the effect of rejuvenation on epigenetic aging.

## High-PRC2 LMRs gain DNA methylation by cell division

Given the correlation of cell replication and the loss of DNA methylation[10], the effect of prolonged cell replication in fully differentiated cells on DNAm gain within high-PRC2 LMRs was assessed using WGBS data from healthy and cancer samples from 8 different cancer types from The Cancer Genome Atlas (TCGA, see ref. 10 for details, Fig. 5a). The general loss of DNAm at CpGs sites genome-wide (Fig. 5a, left panels) was observed, as expected. In addition, similar to the aging methylome, high-PRC2-bound regions gain DNAm in all cancer types. On the other hand, oligodendrocytes, which are post-mitotic in undamaged brains[49], did not show significant DNAm gain in old individuals compared to the young (Fig. 5b). While previous HM450 microarray studies of brain samples demonstrated that PRC2 targets are more likely to gain DNAm compared to non-targets[50], this effect is amplified in mitotic cells. The average DNAm within high-PRC2-binding LMRs in WGBS datasets of passaged primary human fibroblasts showed a very high correlation with passage number (Fig. 5c), providing additional evidence that cell replication of somatic cells highly contributes to the observed gain of methylation.

## PRC2 complex is associated with high-PRC2 LMRs in fully differentiated tissues

We then wondered whether the trends we observed using hESCs PRC2 binding could be replicated by ranking our LMRs based on tissue-specific PRC2 binding. To test this hypothesis, the PRC2-AgeIndex was applied to both neonatal and old fibroblasts, as well as to CD4+ T-cells,

utilizing tissue-specific EZH2 ChIP-Seq data. There was a consistent positive correlation between methylation levels and age or cell passage number (Fig. 6a), mirroring the patterns seen in hESC PRC2 binding (Figs. 2a, 5c, and 6a).

The ranking order of LMRs was then compared between tissue-specific and hESC PRC2 ranking. This comparison revealed a substantial overlap in high-PRC2 LMRs between tissue-specific and hESCs PRC2 ranking (Fig. 6a bottom heatmaps). This large degree of overlap allows hESC PRC2 binding to be a universal reference for the PRC2-AgeIndex.

Interrogating changes in PRC2 binding with age and their role in the increase in methylation at high-PRC2 LMRs, the PRC2-AgeIndex (using hESCs PRC2 ranking) was applied to WGBS data of passaged fibroblast cells from both neonatal and old samples. The expected increase in PRC2-AgeIndex during passaging was observed (Fig. 6b). We then merged the LMRs from neonatal and old samples and applied the PRC2-AgeIndex. In LMRs with high-PRC2 binding (Fig. 6c), not only was there a strong correlation between the number of passages and increased methylation, but also the overall methylation levels were higher in older samples compared to neonatal ones, indicating an age-based separation. Interestingly, when looking at neonatal and old fibroblast EZH2 ChIP-Seq data, EZH2 association with the high-PRC2 LMRs was observed, and no change in binding between neonatal and old samples in these regions was detected (Fig. 6d).

Collectively, the PRC2-AgeIndex, when accounting for tissue-specific EZH2 binding, can effectively distinguish between age and passage number. Additionally, the pattern of PRC2 binding within high-PRC2 LMRs is conserved across various cell types. Furthermore, unchanged EZH2 binding was observed in young and old fibroblasts,

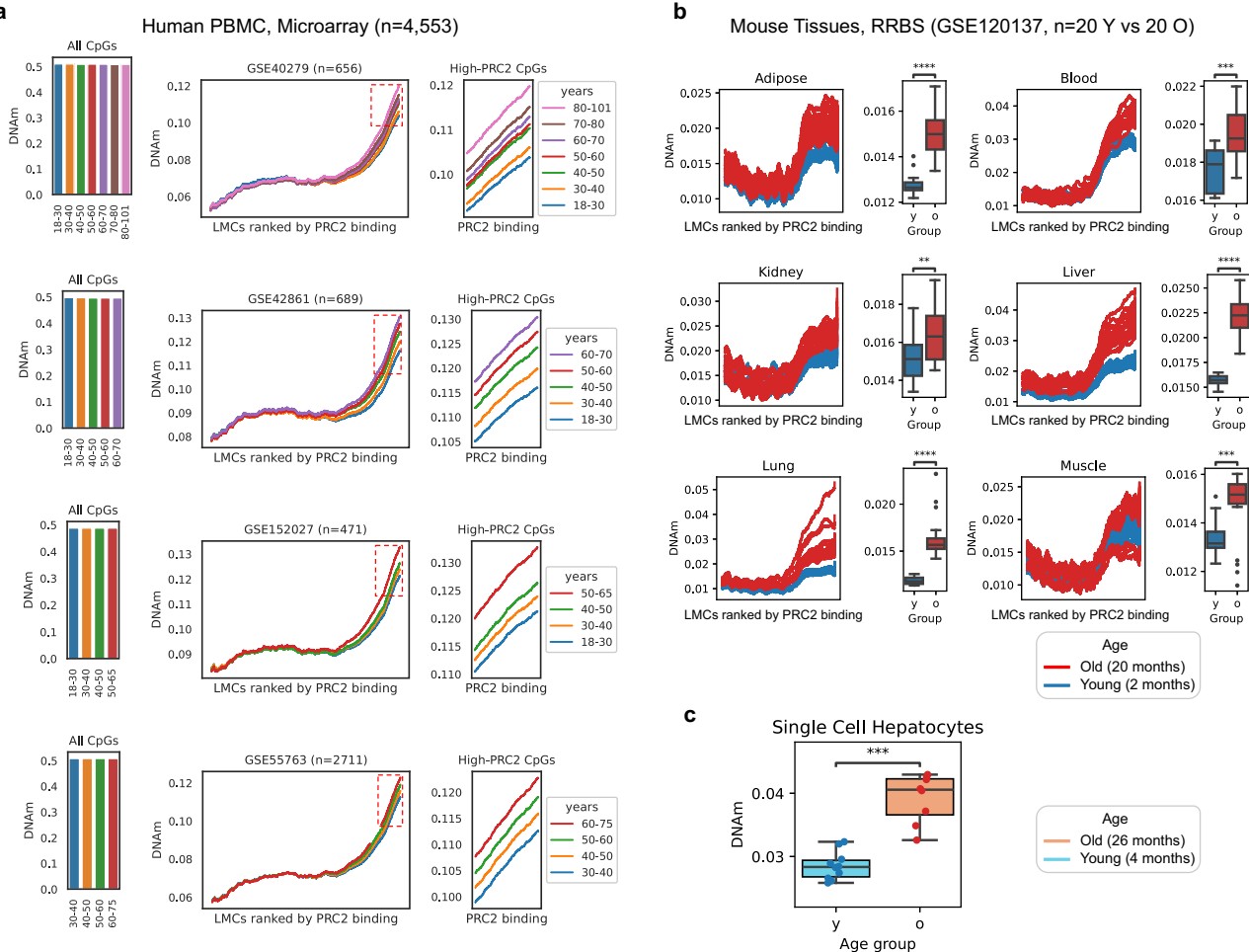

**Fig. 3 | Gain of DNAm at PRC2 targets is measurable using different assays.**
**a** Average DNAm levels at CpGs across the whole genome (left panel) and at low-methylated CpGs (LMCs) ranked by the level of their PRC2 binding in hESCs (right panels). Data were obtained by analyzing DNA methylation microarray data from human PBMC samples. Data from four different HM450 datasets (GSE40279, $n = 656$, GSE42861, $n = 689$, GSE152027, $n = 471$, and GSE55763, $n = 2711$) are shown in each panel. PRC2 binding was estimated by using EZH2 and SUZ12 ChIP-Seq data from ENCODE in the H1 cell line (see "Methods" section). **b** Average DNAm levels in different mouse tissues at LMCs ranked by the level of their PRC2 binding in mESCs (left panel). The right panel shows the same data represented as averages. Data were obtained from an RRBS dataset (GSE120137). One outlier sample was removed

from all analyses (see "Methods" section). One-sided independent $t$-tests were performed between young and old for each tissue (adipose $p = 1.4e{-}09$, blood $p = 2.2e{-}04$, kidney $p = 7.6e{-}03$, liver $p = 1.5e{-}16$, lung $p = 1.1e{-}09$, muscle $p = 4.9e{-}04$). **c** Average DNAm levels at LMCs with high-PRC2 binding in mESCs, in mouse hepatocytes from scWGBS dataset (SRP069120). One outlier sample was removed from each category according to Trapp et al.[74] (see "Methods" section). One-sided independent $t$-tests were performed between young and old samples ($p = 4.9e{-}04$). $p$-value cutoffs for all boxplots are represented graphically by *<0.05, **<0.01, ***<0.001, ****<0.0001. For boxplots, boxes show the quartiles of the dataset with median bar in the center of the box, whiskers represent the range of data (excluding any outliers which are represented as black circles).

---

suggesting that other PRC2-driven epigenetic mechanisms or other epigenetic factors might be responsible for the age-related increase in methylation at PRC2-associated LMRs.

## Discussion

This study takes a significant step forward in addressing two known major challenges in epigenetic studies of aging: the development of a comprehensive genome-wide characterization of the aging methylome and the identification of biologically interpretable epigenetic markers that could be potentially causative of cellular aging[6,51]. Compared to previously developed epigenetic clocks, the PRC2-AgeIndex does not rely on a pre-selected set of individual CpGs, which can be sensitive to perturbation. Indeed, accounting for mean methylation across regions, rather than individual CpGs, has been shown to improve age prediction accuracy and robustness[52]. By taking advantage of multiple methylation-based assays (whole-genome, reduced representation, arrays, and single-cell), the PRC2-AgeIndex robustly discriminates age and cell passaging using the average DNAm of

biologically relevant genomic regions (LMRs), each of them harboring multiple CpGs. These regions were selected based on the specific cellular process of methylation gain at low-methylated regions initially bound by PRC2 in hESCs[25,26,29–33,35–37,53]. Since LMRs are associated with regulatory regions, epigenetic dysregulation through a gain of methylation at these regions with age may have biological significance as confirmed in recent studies[54].

In addition, PRC2-AgeIndex does not rely on "black-box" predictive models that are indeed challenging to interpret. For example, many epigenetic age predictors trained on chronological age alone using penalized regression models include a number of CpGs that do not correlate strongly with age. Such non-age correlated CpGs instead correlate with other cofactors such as smoking, hence their inclusion in the regression model neutralizes their effect (e.g. of smoking) on age prediction. This, in turn, improves chronological age prediction but reduces the biological variability captured[5]. Indeed, the need for having a biologically interpretable biomarker is supported by the evidence that many research groups who previously developed epigenetic

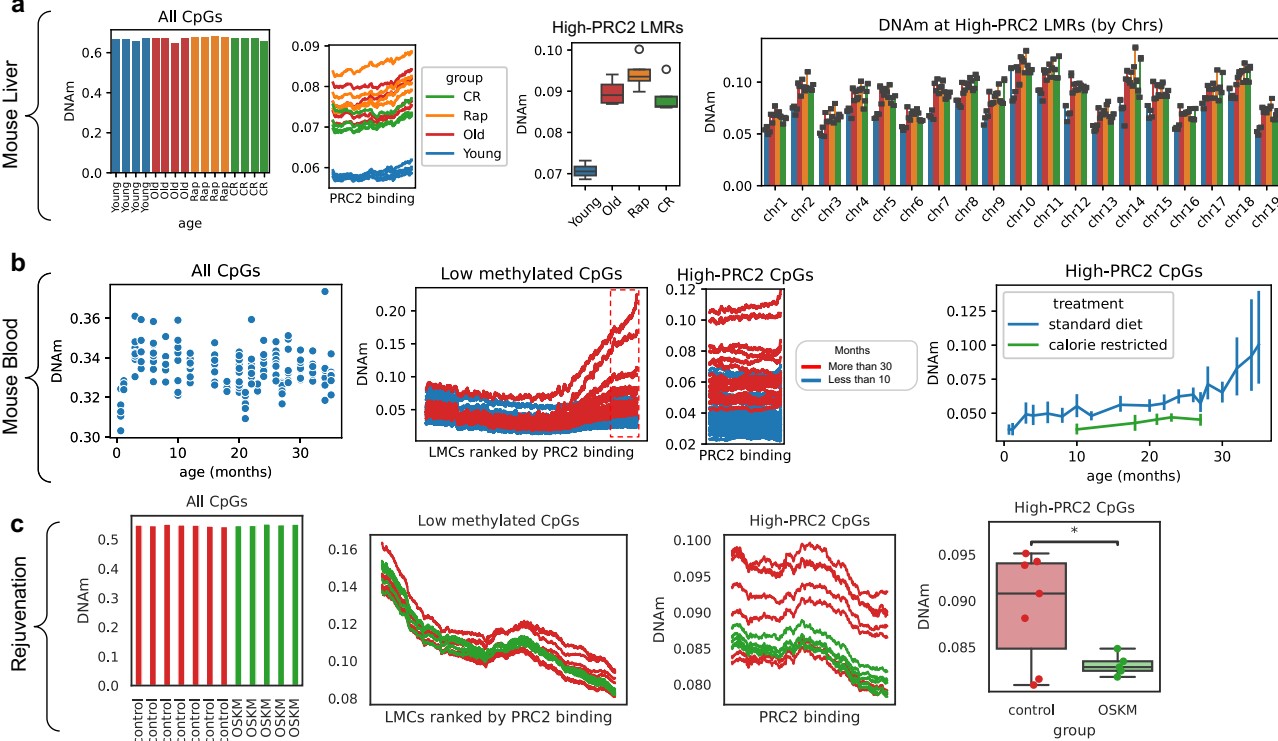

**Fig. 4 | PRC2 targets DNAm as a marker of rejuvenation. a** Average DNAm levels at CpGs across the whole genome (left panel), at the high-PRC2 LMRs (middle panel), in liver samples from young, old, calorie-restricted (CR), and rapamycin-treated (Rap) mice. The boxplot (right), shows the average DNAm levels between the 4 biological replicates (WGBS dataset GSE89274, $n = 16$). **b** Average DNAm levels at CpGs across the whole genome (left panel) and at LMCs ranked by the level of their PRC2 binding in mESCs (middle panels), average DNAm levels in high-PRC2 LMCs at different ages in blood samples from control and calorie-restricted mice (right panel, error bars represent 95% CI for aggregated samples at each timepoint) (RRBS dataset GSE80672, $n = 110$). **c** Average DNAm levels at CpGs across the whole genome (left panel), and at the LMCs ranked by the level of their PRC2 binding in mESCs from long-term partial reprogramming experiments profile with Horvath-MammalianMethylChip40 microarrays (dataset GSE190665, $n = 12$) (middle and left panels; right panel displays difference as a boxplot). One-sided independent $t$-tests were performed between control and OSKM-treated samples ($p = 4.8\text{e}{-}02$), with $p$-value cutoffs represented graphically by *<0.05. For boxplots, boxes show the quartiles of the dataset with median bar in the center of the box, whiskers represent the range of data (excluding any outliers which are represented as black circles).

clocks (that were trained and built to estimate biological age and not rejuvenation) are now developing methods to "de-construct," classify, and/or decipher the clock CpGs[55,56].

The biological implications of the gain of methylation at PRC2-bound regions, together with the putative direct involvement of PRC2 in this process, remain an important and unsolved question whose answer might elucidate the biological significance of the epigenetic basis of aging. Our initial preliminary analysis (Fig. 6b) showed that EZH2 binding does not change between neonatal and old fibroblasts, suggesting that a decline in PRC2 binding might not be directly involved in the age-related increase of methylation at LMRs, however, there is evidence in literature linking PRC2 complex and the methylation of its bound regions[51,57–63]. With the many different theories developing as to the role of PRC2 in aging and other putative cofactors involved in this process, further loss and gain of function experiments are needed to examine the direction and causality of methylation gain at PRC2 targets. Ongoing mechanistic studies of the interactions between PRC2 and target DNA regions, and also between PRC proteins and DNA methyltransferases, will also give further insight into the maintenance of DNAm at PRC2 LMRs and how it might be dysregulated during aging[64].

Another open question is if and how this gain of methylation is affecting the aging outcome. Here, we suggest another potential contributor to the effect. As most PRC2 LMRs contain gene promoters that are silent or lowly expressed in somatic cells, these regions might afford to gain marginal DNAm without being immediately deleterious to cells. However, these gradual age-dependent epigenetic changes might contribute to functional dysregulation in the long-term, associated with aging via the aberrant expression of genes that should be silenced, as previously shown[65]. For example, one theory of aging is that dysregulation of developmental pathways contributes to the aging process[66,67]. A large-scale analysis of 59 tissues amongst 185 mammalian species has revealed conserved CpGs that methylate with age and are enriched at PRC2-binding sites[22]. Consistent with our observations presented in Supplementary Fig. 1e, these regions exhibited an association with developmental genes. This correlation aligns with the fundamental nature of developmental processes, wherein PRC2 exerts a critical influence. The dysregulation of PRC2-associated LMRs/promoters might also involve the formation of R-loops. R-loops have been shown to bind at a subset of PRC targets and are necessary for the repression of gene expression[68,69], hence dysfunction of R-loops might reduce the efficacy of PRC2-mediated repression.

Finally, our results also point to a cell replication-dependent gain of methylation. Future studies in non-cycling cell types such as neurons will be also needed to confirm or deny this hypothesis. Similarly, measuring PRC2-bound LMR methylation in adult stem cells at different states of exhaustion from the same individual could provide insight into the heterogeneity of epigenetics within a single tissue. In summary, we have developed the PRC2-AgeIndex as a robust, versatile, and unbiased biomarker of epigenetic aging as it does not require any prior training or selection of subsets of specific CpGs.

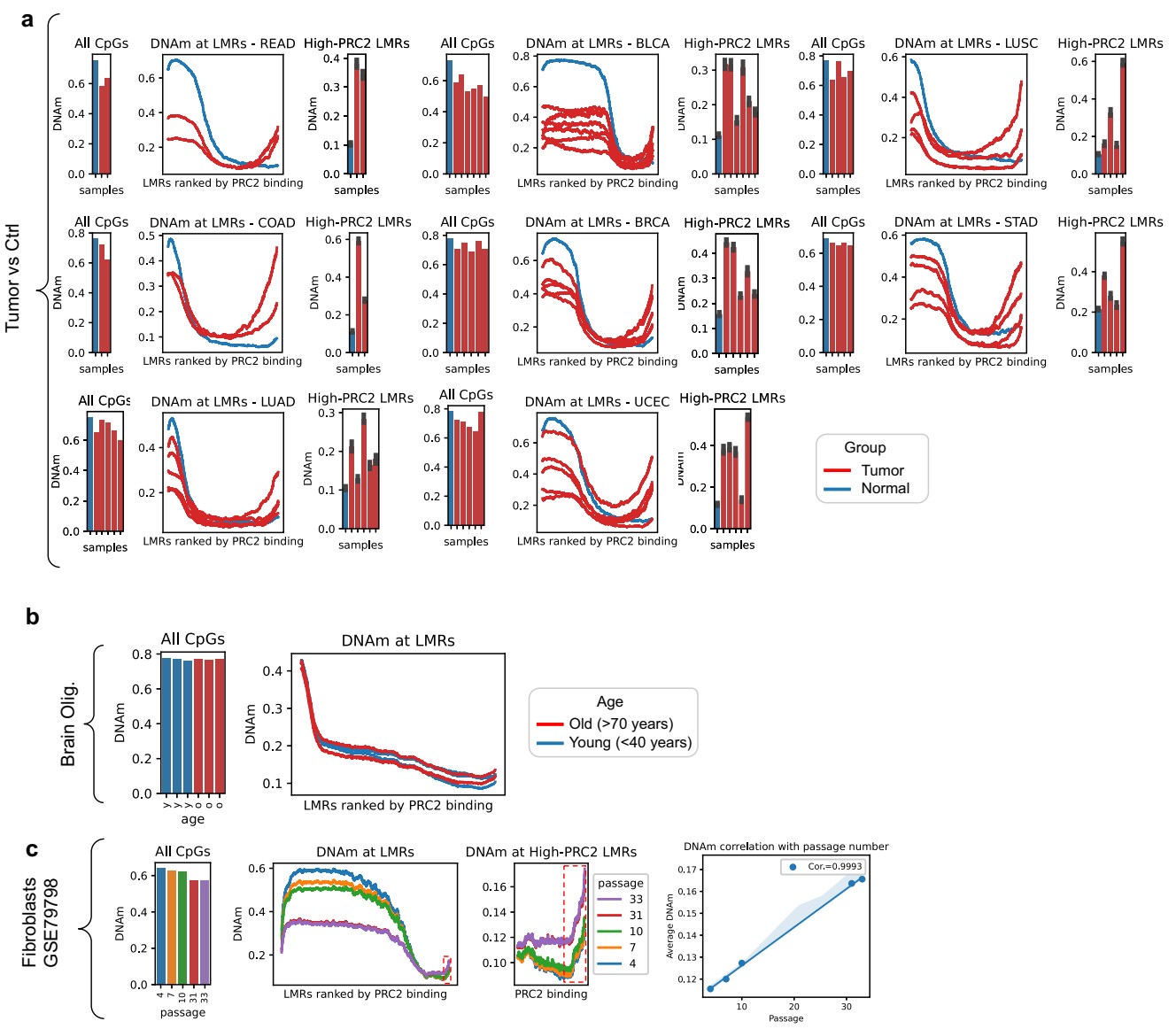

**Fig. 5 | PRC2 targets gain DNA methylation by cell division. a** Average DNAm levels at CpGs across the whole genome (left panel), at LMRs ranked by the level of their PRC2 binding in hESCs (center panel) and within high-PRC2 regions (right panel) from 8 different tumor and normal samples. Data were obtained from TCGA WGBS datasets, (see Supplementary Data 1). Error bars on the right show 95% confidence intervals for average DNAm across high-PRC2 LMRs for each sample. **b** Average DNAm levels at CpGs across the whole genome (left panel) and at LMRs ranked by the level of their PRC2 binding in hESCs (right panel) from young vs old human oligodendrocytes. (WGBS dataset, GSE107729, n = 6). **c** Average DNAm levels at CpGs across the whole genome (left panel) and at LMRs ranked by the level of their PRC2 binding in hESCs (middle panel), correlation between average DNAm at high-PRC2 LMRs and the number of passages (right panel, shaded area of line plot represents 95% CI) in in vitro cultured fibroblasts (GSE79798, n = 5). High-PRC2 LMRs are highlighted with dotted red boxes.

## Methods

### Cell culture conditions

Neonatal fibroblasts were purchased from Cell Applications (#106-05n) and old fibroblasts were isolated from skin tissue collected from deceased healthy individuals. Individuals >80 years old were selected for this study group. All fibroblasts were thawed in passage 2 (P2) and maintained in Human Dermal Fibroblast Media (#116-500, Cell Applications). Media was refreshed every other day and fibroblasts were passaged using Tryple Select (12563011, Gibco) when culture dishes were >80% confluent. Cell pellets were frozen down every passage (P2-P8) to establish a large database of samples. CD4 cells were purified from buffy coats received from Stanford Biobank (Stanford, CA, USA) from three healthy donors aged between 17 and 23. Using RosetteSep Human T-Cell Enrichment Cocktail (STEMCELL Technologies, Cat.n.

15061) T-cells were enriched, CD4 cells were labeled using the CD4 microbeads (Miltenyi Biotec, Cat.n.130-097-048) and purified using LS columns (Miltenyi Biotec, Cat.n.130-042-401).

### WGBS experiments

DNA from neonatal and old fibroblast cells was extracted using the GeneJET genomic DNA extraction kit (Thermo Scientific, Cat.n.K0721). DNA samples were quantified by Qubit 2.0 Fluorometer (Invitrogen, Carlsbad, CA). WGBS library preparation and sequencing reactions were conducted at Novogene Corporation Inc. (Sacramento, CA, USA).

### EZH2 ChIP-Seq experiments

ChIP experiments were performed on chromatin extracts according to the manufacturer's protocol (ChIP-IT High sensitivity, Active

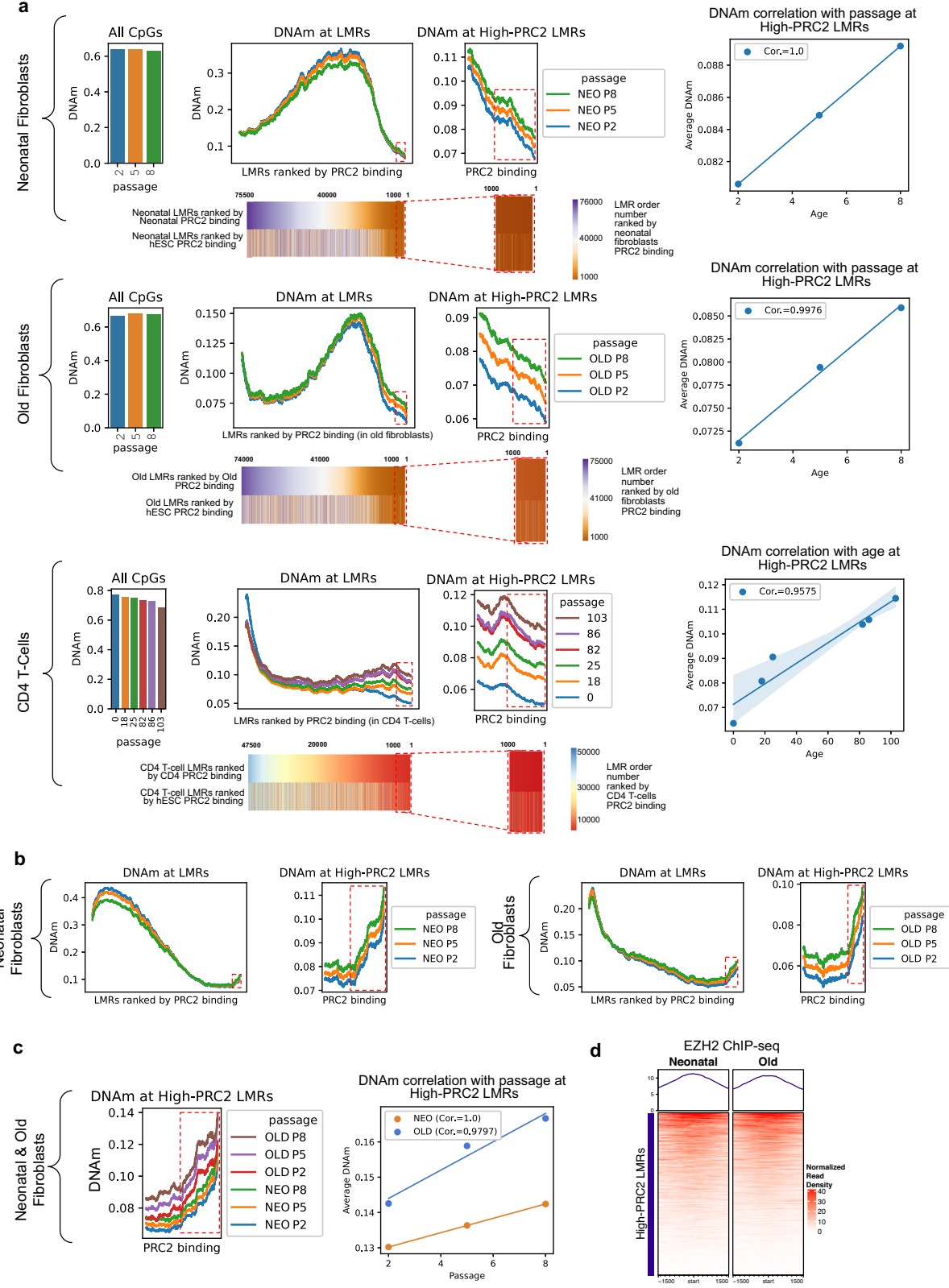

Motif, Cat.n. 53040) and chromatin sheared using the Bioruptor system (Bioruptor Plus, diagenode, Cat.n. B01020001) between 30 and 45 cycles (each cycle 30 s ON/ 30 s OFF high voltage). For the immunoprecipitation, 15 μg of chromatin from CD4 cells of three donors and different passage numbers of neonatal and old fibroblast were used. Sheared chromatin was incubated overnight with 7.5 μl of EZH2 antibody (Ezh2 (D2C9) XP® Rabbit mAb, Cell Signaling, Cat. n. 5246). Immunoprecipitated (IP) and input DNA samples were quantified by Qubit 2.0 Fluorometer (Invitrogen, Carlsbad, CA). ChIP-Seq library preparation and sequencing reactions were conducted at Novogene Corporation Inc. (Sacramento, CA, USA).

**Fig. 6 | PRC2 complex is also associated with high-PRC2 LMRs in fully differentiated tissues. a** Average DNAm levels of neonatal and old passaged fibroblasts, and CD4 T-cells, at CpGs across the whole genome (left panel) and at the LMRs ranked by the level of EZH2-binding data of the same respective tissue (middle panels). Bottom panels show heatmaps of LMR rank number, ordered by EZH2 binding in their respective tissue (top heatmap) and in hESCs (bottom heatmap). Heatmaps are colored by EZH2 binding in the same respective tissue, i.e. purple/orange represents high and low-ranked neonatal/old LMRs ordered by neonatal/old fibroblasts PRC2 binding, respectively, and red/blue represent high and low-ranked CD4 T-cells LMRs ordered by CD4 T-cells PRC2 binding, respectively. Right panel shows the correlation of mean methylation of high-PRC2 LMRs against age or passage number. **b** Average methylation levels at LMRs ranked by PRC2 binding in hESCs for neonatal fibroblasts (left panels) and old fibroblasts (right panels).

**c** Average DNAm levels of LMRs calculated from neonatal and old samples merged (all passages), ranked by PRC2 binding in hESCs (left panel) and correlation between methylation and age for high-PRC2 LMRs of both neonatal and old in vitro passaged fibroblasts (right panel). **d** Heatmap of normalized read density of the high-PRC2 neonatal/old LMRs in neonatal and old fibroblasts (passage 2). WGBS Neonatal and old fibroblasts were generated by our lab (GSE253987, $n = 3$ for neonatal samples (one donor) and $n = 3$ for old samples (one donor)), CD4 T-cell samples are the same as used in Fig. 2 (6 samples 18–86 years old, GSE79798 and 2 samples 0 and 103 years old, GSE31263). ChIP data for fibroblast and CD4 T-cells were generated by our lab (GSE253987, two donors pooled at passage 2 for neonatal fibroblasts, three donors pooled at passage 2 for old fibroblasts, three donors pooled for CD4 T-cells). High-PRC2 LMRs are highlighted with dotted red boxes. Line plots with >3 samples plotted have a shaded area representing 95% CI.

## WGBS and RRBS data processing
Whole genome and reduced representation bisulfite sequencing reads were downloaded from the Sequence Read Archive (SRA) (see Source Data for a list of dataset accession numbers). Neonatal and old passaged fibroblast samples and CD4+ T-cell samples generated in our lab were whole-genome enzymatic methyl-sequenced by Novogene. Reads were preprocessed by removing adapters using (*Babraham Bioinformatics - Trim Galore!*) version 0.6.7 and subsequently mapped to their corresponding reference genome (either hg38 or mm10) using *abismal* version 2.0.0[70]. The enzymatic methyl-sequenced samples were mapped with bismark version 2.5.1 as recommended by Novogene and the resulting BAM files were converted to dnmtools/methpipe formatted SAM files using dnmtools 1.4.1. The output SAM files were analyzed using methpipe version 5.0.1[71]. Paired-end reads were merged using the *format_reads* tool, after which reads were sorted by chromosome coordinates using *samtools sort*[72]. PCR duplicates, defined as identical reads that map to the same location, were removed using the *duplicate-remover* tool in methpipe. Individual CpG methylation levels were quantified from SAM files using *methcounts* and summary statistics on CpG coverage, depth, and average methylation were quantified using the *levels* program. Methylation levels of symmetric CpGs in opposite strands of the reference were merged using the *symmetric-cpgs* program.

## PRC2-AgeIndex generation
Low-methylated regions (LMRs) were identified from symmetric CpG methylation levels using the *hmr* program with default parameters. For LMRs generated in Fig. 6c, neonatal and old samples from all passages were merged using dnmtools merge function, and LMRs were generated from the resulting merged.meth file. UCSC genome browser tracks for symmetric CpGs were created using the *wigToBigWig* program. LMRs were ranked based on the average PRC2-binding levels (see ChIP-Seq section), and blacklist regions were filtered[73]. The horizontal axis in all LMR figures shows ranked PRC2 regions from lowest binding (left) to highest binding (right). In order to reduce the noise and visualize the trends in DNAm changes, sliding windows and smoothing are used and the average DNAm is calculated and displayed within each sliding window.

## Methylation array analysis
All methylation array datasets were downloaded from GEO (GSE190665, GSE152027, GSE40279, GSE42861, and GSE55763). No additional normalization of methylation levels was performed.

## Binning
In order to reduce the noise and visualize the trends in DNAm changes, sliding windows and smoothing are used and the average DNAm is calculated and displayed within each sliding window.

## Outliers
Among the 8 human WGBS CD4+ T samples, one adult sample was removed since its overall methylation was outside the range between the neonatal and centenarian samples. Among the 40 mouse RRBS samples, one mouse had significantly higher methylation levels across different tissues and was removed from the analyses. Among scWGBS hepatocyte samples, following the analyses by Trapp. et al.[74] and Levine et al.[55], one sample was removed from each of the young and old groups. A list of all datasets with outliers is available in Supplementary Data 1. In addition, scWGBS analysis was limited to regions of high CpG densities to compensate for the sparsity of covered CpGs across different cells.

## Public ChIP-Seq data
EZH2 and SUZ12-binding regions and levels were extracted from ENCODE ChIP-Seq data from H1 human ESC cell line publicly available under accession numbers ENCSR000ASY and ENCSR000ATS and from mouse embryonic cells lines (E14) publicly available under accession numbers GSM2472741 and GSM3243624. PRC2-bound regions are identified using fold change binding levels from the Encode pipeline[75]. "High-PRC2 LMRs" for each tissue were selected based on the top 1000 LMRs with the lowest average *p*-values from EZH2 and SUZ12 Chip-Seq data. Histogram of LMR size in bp was plotted in R-4.2.2.

## EZH2 ChIP-Seq processing
The following processing steps were conducted as per the ENCODE3 ChIP-Seq Pipeline. CD4 T-cells (three donors, aged 23, 17, and 17) and passage 2 neonatal and old fibroblast samples were trimmed using Trimgalore! (see "Methods" section - WGBS and RRBS data) and paired-end mapped to hg38 using Bowtie2 version 2.5.0 (max fragment length for valid paired-end alignments set to 2000). Output SAM files were filtered to keep matching, mapped pairs only with a MAPQ > 30, and duplicate reads removed, using samtools version 1.18 and Picard Tools version 3.0.0. MACS2 (version 2.2.7.1) callpeak was run on each individual tissue with replicates merged for each tissue/passage, and a fold enrichment signal track was built using the bdgcmp (narrowpeak files were also called for each replicate separately for differential binding analysis, see next section). The resulting bedgraph files were sorted, filtered for autosomes, and converted to bigwig files (using the bedGraphToBigWig function version 2.9) and applied to the PRC2-AgeIndex as per "Public ChIP-Seq" and "PRC2-AgeIndex generation" sections.

## Plotting EZH2 binding in high-PRC2 LMRs
Mapped BAM files (converted from SAM files using samtools) and narrowpeak files generated with MACS2 (see previous paragraph) for each replicate were then used for differential binding analysis between neonatal and old samples at passage 2, using DiffBind version 3.8.4 in R. Greylist and blacklist regions were filtered using the dba.blacklist function, and counts were read and normalized using dba.normalize, both from the DiffBind package. The dba.plotProfile function from DiffBind was used to generate Fig. 6d for the top 1000 PRC2-bound neonatal and old LMRs.

## LMR ordered heatmaps

To create the hESC/tissue LMR ordered heatmap in Fig. 6a, first, the order of the same tissue PRC2 binding was queried against the corresponding tissue LMRs sorted by hESC PRC2 binding using the findOverlaps function from the IRanges package (version 2.32.0). The numbered list of LMRs was then plotted using the pheatmap package (version 1.0.12).

## Comparison of epigenetic clocks with the PRC2-AgeIndex

To effectively compare the effect sizes of the PRC2-AgeIndex with published DNAm-based epigenetic clocks, clock sites were extracted from the T-cell, epidermis and fibroblast WGBS datasets (from Figs. 2a and 5c) using coordinate data stored in the Illumina Array Manifest table. Public epigenetic clocks[39] were then applied using Biolearn[38] and the Pearson correlation for each clock was plotted.

## Characterization of LMRs

The following analysis in this section was conducted using R-4.2.2 unless otherwise stated. Genomic annotation of LMRs was conducted using the annotatr package (ver. 1.24.0) (annotatr -Bioconductor) using accessions from txdb USCS (package TxDb.Hsapiens. UCSC.hg38.knownGene ver. 3.16.0 (TxDb.Hsapiens.UCSC.hg38.known Gene -Bioconductor), TxDb.Mmusculus.UCSC.mm10knownGene ver. 3.10.0 (TxDb.Mmusculus.UCSC.mm10.knownGene - Bioconductor)), org.Hs.eg.db (ver. 3.16.0) (org.Hs.eg.db - Bioconductor) and org.Mm.eg.db (ver. 3.16.0) (org.Mm.eg.db - Bioconductor). Repeat annotations where two annotations share the same coordinates (e.g. two exons with identical coordinates) were removed. The total bp of individual genomic annotations (all "Genic" and "Regulatory/Noncoding" terms in Supplementary Fig. 1c) for the top 1000 PRC2-binding LMRs in a given tissue were divided by total bp of all genomic annotations in the top 1000 PRC2-binding LMRs of the respective tissue. The corresponding proportion of a genomic feature was also calculated in the same manner for all LMRs in a given tissue (to use as genomic background to check against the top 1000 LMR features). The top 1000 PRC2-binding LMR proportion of a given annotation was then divided by the corresponding feature proportion in all LMRs of the same tissue, and then the log2 of the result was calculated. The resulting log2 fold change was plotted using ggplot2 (ver. 3.4.0) in Supplementary Fig. 1d. The same process was repeated for CpG features and plotted in Supplementary Fig. 1d. The summarize_catagorical function in annotatr was used to summarize the features and genes present in each LMR to produce an annotated list of top 1000 LMRs for CD4 T-cells, epidermis, fibroblasts, and mouse liver (see Source Data File 1). For each tissue analyzed, unique promoters, enhancers, and first exons were extracted from the top 1000 PRC2-binding LMRs and submitted on GREAT[76] for Gene Ontology analysis. Annotation for the two nearest genes within 1000 kb per feature was selected and "include curated regulatory domains" was checked. The unique promoter, enhancer, and first exon annotations for all LMRs in a corresponding tissue were used as the background for comparison. The top 20 (in order of fold enrichment) of "GO Biological Processes" where hypergeometric $p$-value = 0 were selected and plotted using ggplot2 in Supplementary Fig. 1c. Overlapping CpGs between LMRs was assessed by annotating (using the annotatr package) the top 1000 PRC2-binding LMRs with individual CpGs from BSgenome.Hsapiens.UCSC.hg38 (ver. 1.4.4 BSgenome.Hsapiens.UCSC.hg38 - Bioconductor). The overlaps of CpGs between datasets were plotted for Supplementary Fig. 1b using ChIPpeakAnno (ver. 3.32.0 ChIPpeakAnno - Bioconductor) and Vennerable (ver. 3.1.0.9000).

## Statistical analyses

Statistical analyses in Figs. 3 and 4 are based on simple one-sided independent $t$-tests.

## Reporting summary

Further information on research design is available in the Nature Portfolio Reporting Summary linked to this article.

## Data availability

The Fibroblast WGBS, ChIP-Seq data, and CD4 T-cell ChIP-Seq data generated in this study have been deposited in the in Gene Expression Omnibus (GEO) database under accession code GSE253987. The public data analyzed in this study are referenced in the Supplementary Data 1. Source data are provided with this paper.

## Code availability

Our scripts to analyze the WGBS data, produce the figures, and generate the final PRC2 LMRs are freely accessible on GitHub at https://github.com/moqri/PRC2-AgeIndex.

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

## Acknowledgements

V.S. is supported by the MCHRI Woods Family Endowed Scholarship in Pediatric Translational Medicine (Stanford Maternal & Child Health Research Institute), the Breakthrough in Gerontology Award (BIG Award, AFAR/Glenn Foundation), and the Milky Way Research Foundation. A.C. is supported by the DiGenova Postdoc Seed Grant (Stanford University). M.M. is funded by NIH T15 in Biomedical Informatics and NIH T32 Aging Research at Stanford. M.P.S. is supported by the CEGS 5RM1HG00773509. We would like to express our deepest gratitude to the members of the Sebastiano and Snyder Labs for their invaluable contributions and unwavering support throughout the course of this research. Special thanks to our dedicated technician, whose hard work was instrumental in the success and facilitation of this project. A special thank you goes to all the individuals and facilities that provided the necessary resources and assistance to make this research possible. Your collective efforts and commitment have been crucial in bringing this study to fruition.

## Author contributions

M.M. and A.C. designed and conceived, the study. D.J.S. provided a major contribution in performing the additional analyses during the revision of the manuscript. A.C. performed and supervised the experimental part, with the help of S.R. and T.A.J. M.M. performed and supervised the bioinformatic analyses. T.M., D.N., G.S.B., K.Y., A.T., K.A.A., E.O., A.S., and W.Z. helped with additional analyses of DNA Methylation data and provided feedback on the analysis. C.M., V.G., S.H., M.P.S., and V.S. provided guidance to the experiments or the analysis. The paper was written by M.M. and A.C. with major contributions from D.J.S., and suggestions from all authors. M.P.S. and V.S. supervised the entire project and the preparation of the manuscript.

## Competing interests

V.S. is a co-founder, SAB Chairman, Head of Research, and shareholder of Turn Biotechnologies. S.H. is a founder of the non-profit Epigenetic Clock Development Foundation. M.M., A.C., V.S., M.P.S., and V.N.G. have filed patents on measuring aging. V.N.G. is supported by NIA grants. The remaining authors declare no competing interests.
