## [Peer Review File · Nature Communications]

PRC2-AgeIndex as a universal biomarker of aging and rejuvenationEditorial note: This manuscript has been previously reviewed at another journal that is not operating a transparent peer review scheme. This document only contains reviewer comments and rebuttal letters for versions considered at *Nature Communications*.

REVIEWER COMMENTS

Reviewer #1 (Remarks to the Author):

The authors have attempted to address my main concern by comparing their Index to existing ones. However, they did so only in the context of chronological age prediction, which is not technically the goal of aging biomarkers. That said, I am fine with the paper as it stands and additional interrogation of the validity of the presented Index will likely be assessed over time as groups begin to apply it. I would however, suggest the authors be more conservative in their claims as I still do not feel they have actually demonstrated their measure has greater utility than existing ones when it comes to estimating biological (rather than chronological) age.

Reviewer #4 (Remarks to the Author):

The manuscript "PRC2-AgeIndex: a universal biomarker of aging and rejuvenation" presents an observation that PRC2-bound low-methylated regions gain methylation with age, that these regions account for 90% of methylation gain with age, and that they can be viewed as a pan-tissue human and mouse biomarker of aging. These findings are not new, and various aspects of them have been reported previously with correct citations. The current manuscript appears to focus on a method utilizing those observations for age prediction. PRC2-AgeIndex was benchmarked against 41 existing clocks, outperforming most for age prediction. An interesting observation is that, in contrast to the general loss of methylation with age, low-methylated regions gain methylation during cycles of cell division.

The manuscript leaves mixed feelings because its message and deliverables are unclear. Is it a computational analysis? With experimental validation? A new method? Implemented as a tool? There are minor inconsistencies, especially in the revised parts, that leave an

impression of rushed work.

- I.82. The current manuscript focuses on a method utilizing those observations for age prediction. But there is no actual method or tool that others can use. The code at <https://github.com/moqri/PRC2-AgeIndex> lacks documentation, any guidance how to use PRC2-AgeIndex. Table S2 is the only practical component providing coordinates of the low-methylated regions. Implementation of the method in a useful R or Python code utilizing those coordinates is warranted.

- It appears, the manuscript generated new omics data. Why there's no reviewer's access to the project PRJNA804539?

- Methods description sometimes broad and imprecise. E.g., "Greylists and blacklist regions were filtered" - which ones, what are data sources, especially for Greylists? Missing references, e.g., "Biolearn (ref)"

- I.162. What is "WGBA"? On line 167, there's another variant, WGBS.

- Figure S1b - "Venn diagram of overlapping CpGs between top 1000 PRC2-binding LMRs.." - the numbers on the Venn diagram are much larger than 1K.

- Figure S1c - "Log2 fold change", of what?

- Figure S2 - the legend refers to panels a, b, and c, but the letters are missing from the figure.

- I.529. - references to non-existent Figure S2e

- I.531. - What is "S.Table 2"

Reviewer #4 (Remarks on code availability):

Undocumented, poorly commented, not a method but a script dump.

Reviewer #1 (Remarks to the Author):

The authors have attempted to address my main concern by comparing their Index to existing ones. However, they did so only in the context of chronological age prediction, which is not technically the goal of aging biomarkers. That said, I am fine with the paper as it stands and additional interrogation of the validity of the presented Index will likely be assessed over time as groups begin to apply it. I would however, suggest the authors be more conservative in their claims as I still do not feel they have actually demonstrated their measure has greater utility than existing ones when it comes to estimating biological (rather than chronological) age.

We thank reviewer 1 for the feedback throughout the revision process which has greatly improved the manuscript. We agree that further studies utilizing the Index will provide further insights.

Reviewer #4 (Remarks to the Author):

The manuscript "PRC2-AgeIndex: a universal biomarker of aging and rejuvenation" presents an observation that PRC2-bound low-methylated regions gain methylation with age, that these regions account for 90% of methylation gain with age, and that they can be viewed as a pan-tissue human and mouse biomarker of aging. These findings are not new, and various aspects of them have been reported previously with correct citations. The current manuscript appears to focus on a method utilizing those observations for age prediction. PRC2-AgeIndex was benchmarked against 41 existing clocks, outperforming most for age prediction. An interesting observation is that, in contrast to the general loss of methylation with age, low-methylated regions gain methylation during cycles of cell division. The manuscript leaves mixed feelings because its message and deliverables are unclear. Is it a computational analysis? With experimental validation? A new method? Implemented as a tool? There are minor inconsistencies, especially in the revised parts, that leave an impression of rushed work.

- l.82. The current manuscript focuses on a method utilizing those observations for age prediction. But there is no actual method or tool that others can use. The code at <https://github.com/moqri/PRC2-AgeIndex> lacks documentation, any guidance how to use PRC2-AgeIndex. Table S2 is the only practical component providing coordinates of the low-methylated regions. Implementation of the method in a useful R or Python code utilizing those coordinates is warranted.

We thank the reviewer for the valuable feedback. We have rephrased line 82 based on the feedback. The GitHub page has now been updated with two directories, one with all the scripts needed to generate the manuscript figures (with input files too big to store on GitHub, but available on request). The second folder contains a user-friendly example with instructions for recreating Fig 6b; neonatal passaged fibroblasts ordered by hESC PRC2 binding.

- It appears, the manuscript generated new omics data. Why there's no reviewer's access to the project PRJNA804539?

We apologize for missing this information in the text, but our accession number has been updated to **GSE253987** and is now available to the reviewer with the following key: **qxgfmaqghxebvqz** and will be public upon publication. Thank you for having noticed and pointed that out.

- Methods description sometimes broad and imprecise. E.g., "Greylists and blacklist regions were filtered" - which ones, what are data sources, especially for Greylists? Missing references, e.g., "Biolearn (ref)"

We thank the reviewer for having noticed this imprecision; Both greylists and blacklists were filtered using the dba.blacklist function regarding the Diffbind analysis, which has been corrected in lines 498-500. A reference to the ENCODE blacklist regions used when generating the PRC2 binding of LMRs was also added to line 455. Biolearn and Clockbase references were also added to line 510.

- l.162. What is "WGBA"? On line 167, there's another variant, WGBS.

"WGBA" has been corrected to "WGBS", thank you for having noticed it.

- Figure S1b - "Venn diagram of overlapping CpGs between top 1000 PRC2-binding LMRs.." - the numbers on the Venn diagram are much larger than 1K.

Overlapping CpGs between LMRs were checked, as mentioned in the title of the figure. This is because multiple LMRs in one tissue can be contained within one LMR in another tissue, which can lead to misleading results. Hence, we found it more representative to present the number of CpGs that overlap between the top 1000 LMRs.

- Figure S1c - "Log2 fold change", of what?

The log2 fold change in figure S1C, represents the enrichment of the genomic annotation proportions in top 1000 PRC2-binding LMRs compared to respective genomic proportions in all LMRs (used as background) for a given tissue, as explained in the figure legend and lines 522-529 in the methods. We have rephrased the methods section (lines 522-529) to make it clearer and updated the figure y axis to be more descriptive. Thank you for the feedback.

- Figure S2 - the legend refers to panels a, b, and c, but the letters are missing from the figure.

Apologies, the letters are present in our version of the supplementary file. We will double check before resubmitting, thank you.

- l.529. - references to non-existent Figure S2e

Thank you for spotting this; it has been removed.

- l.531. - What is "S.Table 2"

More explanation has been given to Table S2, (Line 531-532) and the legend has been added to the table file. Thank you for having noticed it.

Reviewer #4 (Remarks on code availability):

Undocumented, poorly commented, not a method but a script dump.

We thank the reviewer for this feedback and we apology for this, we have extensively improved the code and generated a new version available at <https://github.com/moqri/PRC2-AgeIndex>.

REVIEWERS' COMMENTS

Reviewer #4 (Remarks to the Author):

The limited code documentation and the lack of an applicable method remain the drawbacks. To address the lack of documentation in <https://github.com/moqri/PRC2-AgeIndex>, two folders were created. Naming the folder "user friendly" does not make it as such. There's no instructions about prerequisites, Python version, installation of additional tools. Download of input files, especially public, could be automated and it is not. It does not appear a self-standing example, just a folder with the code for a specific figure.

Reviewer #4 (Remarks on code availability):

The limited code documentation and the lack of an applicable method remain the drawbacks. To address the lack of documentation in <https://github.com/moqri/PRC2-AgeIndex>, two folders were created. Naming the folder "user friendly" does not make it as such. There's no instructions about prerequisites, Python version, installation of additional tools. Download of input files, especially public, could be automated and it is not. It does not appear a self-standing example, just a folder with the code for a specific figure.

Reviewer #4 (Remarks to the Author):

"The limited code documentation and the lack of an applicable method remain the drawbacks. To address the lack of documentation in <https://github.com/moqri/PRC2-AgeIndex>, two folders were created. Naming the folder "user friendly" does not make it as such. There's no instructions about prerequisites, Python version, installation of additional tools. Download of input files, especially public, could be automated and it is not. It does not appear a self-standing example, just a folder with the code for a specific figure."

Response:

We thank the reviewer for the feedback. We have added package versions and python version to the user-friendly example on Github. We have also added functionality to download the required files from GEO to run the example, as well as the means to download the ENCODE ChIP files required to sort LMRs by PRC2 binding. We have also made clear in the example code and README that the methylation bigwig files downloaded can be substituted for any other methylation bigwig files that the user desires, so that the user can apply the PRC2-AgeIndex to their own desired samples.